

# Partitioning climate projection uncertainty with multiple Large Ensembles and CMIP5/6

Flavio Lehner[1,2], Clara Deser[2], Nicola Maher[3], Jochem Marotzke[3], Erich Fischer[1], Lukas Brunner[1], Reto Knutti[1], Ed Hawkins[4]

[1]Institute for Atmospheric and Climate Science, ETH Zürich, Zürich, Switzerland
[2]Climate and Global Dynamics Laboratory, National Center for Atmospheric Research, Boulder, USA
[3]Max Planck Institute for Meteorology, Hamburg, Germany
[4]National Centre for Atmospheric Science, Dept. of Meteorology, University of Reading, Reading, UK

*Correspondence to*: Flavio Lehner (flehner@ucar.edu)

**Abstract.** Partitioning uncertainty in projections of future climate change into contributions from internal variability, model response uncertainty, and emissions scenarios has historically relied on making assumptions about forced changes in the mean

and variability. With the advent of multiple Single-Model Initial-Condition Large Ensembles (SMILEs), these assumptions can be scrutinized, as they allow a more robust separation between sources of uncertainty. Here, the iconic framework from Hawkins and Sutton (2009) for uncertainty partitioning is revisited for temperature and precipitation projections using seven SMILEs and the Climate Model Intercomparison Projects CMIP5 and CMIP6 archives. The original approach is shown to work well at global scales (potential method error <20%), while at local to regional scales such as British Isles temperature or

Sahel precipitation, there is a notable potential method error (up to 50%) and more accurate partitioning of uncertainty is achieved through the use of SMILEs. Whenever internal variability and forced changes therein are important, the need to evaluate and improve the representation of variability in models is evident. The available SMILEs are shown to be a good representation of the CMIP5 model diversity in many situations, making them a useful tool for interpreting CMIP5. CMIP6 often shows larger absolute and relative model uncertainty than CMIP5, although part of this difference can be reconciled with

the higher average transient climate response in CMIP6. This study demonstrates the added value of a collection of SMILEs for quantifying and diagnosing uncertainty in climate projections.

## 1 Introduction

Climate change projections are uncertain. Characterizing this uncertainty has been helpful for scientific interpretation and

guiding model development, but also for science communication. With the advent of Coupled Model Intercomparison Projects



(CMIPs), a systematic characterization of projection uncertainty became possible, as a number of climate models of similar complexity provided simulations over a consistent time period and with the same set of emissions scenarios. Uncertainties in climate change projections can be attributed to different sources - in context of CMIP to three specific ones (Hawkins and Sutton 2009):

-    Uncertainty from internal unforced variability, i.e., the fact that a projection of climate is uncertain at any given point in the future due to the chaotic and thus unpredictable evolution of the climate system. This uncertainty is inherently irreducible on time scales after which initial condition information has been lost (typically a few years or less for the atmosphere). Internal variability of a climate model can be best estimated from a long control simulation or a large ensemble, including how variability might change under external forcing (Brown et al. 2017; Maher et al. 2018).

-    Climate response uncertainty (hereafter "model uncertainty", for consistency with historical terminology), i.e., structural differences between models and how they respond to external forcing. Arising from choices made by individual modeling centers during the construction and tuning of their model (e.g., parameterizations of unresolved processes), this uncertainty is in principle reducible as the differences between models (and between models and observations) are artifacts of model imperfection. To be able to distinguish model uncertainty from internal variability

uncertainty, a robust estimate of a model's "forced response", i.e., its response to external radiative forcing of a given emissions scenario, is required. Again, a convenient way to obtain a robust estimate of the forced response is to average over a large initial-condition ensemble from a single model (Deser et al. 2012; Frankcombe et al. 2018; Maher et al. 2019).

      -    Radiative forcing uncertainty (hereafter "scenario uncertainty"), i.e., lack of knowledge of future radiative forcing

that arises primarily from unknown future greenhouse gas emissions. Scenario uncertainty can be quantified by comparing a consistent and sufficiently large set of models run under different emissions scenarios. This uncertainty is considered irreducible from a climate science perspective, as the scenarios are socio-economic 'what-if' scenarios and do not have any probabilities assigned (which does not imply they are equally likely in reality).

In a landmark paper, Hawkins and Sutton (2009; hereafter HS09) made use of the most comprehensive CMIP archive at the time (CMIP3; (Meehl et al. 2007)) to perform a separation of uncertainty sources for surface air temperature at global to regional scales. Due to the lack of large ensembles or even multiple ensemble members from individual models in CMIP3, it was necessary to make an assumption about the forced response of a given model. In HS09, a 4[th] order polynomial fit to global and regional temperature time series represented the forced response, while the residual from this fit represented the internal

variability. Using 15 models and 3 emissions scenarios, this enabled a separation of sources of uncertainty for temperature projections, which was later expanded to precipitation ((Hawkins and Sutton 2011); hereafter HS11). While other important research into uncertainty quantification was ongoing (Knutti and Sedláček 2012; Rowell 2012), it was probably the accessible and illustrative figures in HS09 that created a powerful narrative of reducible and irreducible uncertainties in climate projections.






However, the HS09 approach is likely to conflate internal variability with the forced response in cases where there exists low-frequency (decadal-to-multi-decadal) internal variability, after large volcanic eruptions, or when the forced signal is weak, making the statistical fit a poor estimate of the forced response (Kumar and Ganguly 2018). HS09 tried to circumvent this issue by focusing on large enough regions and a future without volcanic eruptions, such that there was reason to believe that
the spatial averaging would dampen variability sufficiently for it not to alias into the estimate of the forced response described by the statistical fit.

An alternative to statistical fits to estimate the forced response in a single simulation is a Single-Model Initial-Condition Large Ensemble (SMILE). A SMILE enables the robust quantification of a model's forced response and internal variability via
computation of ensemble statistics, provided the ensemble size is large enough. Due to their computational costs, SMILEs have not been wide-spread even in the latest CMIP6 archive. Nevertheless, since HS09, a number of modeling centers have conducted SMILEs (Selten et al. 2004; Deser et al. 2012; Kay et al. 2015; Deser et al. in review, and references therein). Thanks to their sample size, SMILEs have been applied most successfully to problems of regional detection and attribution (Deser et al. 2012; Sanderson et al. 2015; Frölicher et al. 2016; Rodgers et al. 2015; Mankin and Diffenbaugh 2015; Lehner et
al. 2017a, 2018; Kumar and Ganguly 2018; Schlunegger et al. 2019; Marotzke 2019), extreme and compound events (Fischer et al. 2014, 2018; Schaller et al. 2018; Kirchmeier-Young et al. 2017), and as testbeds for method development (Lehner et al. 2017b; McKinnon et al. 2017; Frankignoul et al. 2017; Wills et al. 2018; Sippel et al. 2019; Barnes et al. 2019).

The availability of a collection of SMILEs (Deser et al. in review) now provides the ability to scrutinize and ultimately drop
the assumptions of the original HS09 approach. Further, it allows for a separation of the sources of projection uncertainty at smaller scales and for noisier variables. With multiple SMILEs, one can directly quantify the evolving fractional contributions of internal variability and models' structural differences to the total projection uncertainty under a given emissions scenario. A SMILE gives a robust estimate of a model's internal variability and multiple SMILEs thus enable differentiating robustly between magnitude of internal variability across models. Maher et al. (in review) used multiple SMILEs to show that the
magnitude of internal variability differs between models to the point that it affects whether internal variability or model uncertainty is the dominant source of uncertainty for 30-year temperature projections. Building on that, one can also assess the contribution of any forced change in internal variability by comparing the time-evolving variability across ensembles members with the constant variability from present-day or a control simulation (Pendergrass et al. 2017; Maher et al. 2018). Here, we revisit the HS09 approach using temperature and precipitation projections from multiple SMILEs, CMIP5 and CMIP6 to
illustrate where it works, where it has limitations, and how SMILEs can be used to complement the original approach.



## 2. Data and Methods

### 2.1 Model simulations

We make use of seven publicly available SMILEs that are part of the Multi-Model Large Ensemble Archive (MMLEA; Table
1), centrally archived at the National Center for Atmospheric Research (Deser et al. in review). All use CMIP5-class models
(except MPI, which is closer to its CMIP6 version), although not all of the simulations were part of the CMIP5 submission of
the individual modelling centers and were thus not accessible in a centralized fashion until recently. All SMILEs used here
were run under the standard CMIP5 'historical' and Representative Concentration Pathway 8.5 (RCP8.5) forcing protocols
and are thus directly comparable to corresponding CMIP5 simulations (Taylor et al. 2007). The models range from ~2.8° to
~1° horizontal resolution and from 16 to 100 ensemble members. For model evaluation and other applications, the reader is
referred to the references in Table 1. We also use all CMIP5 models for which simulations under RCP2.6, RCP4.5 and RCP8.5
are available (28; Supplementary Information Table S1) and all CMIP6 models for which simulations under SSP1-2.6, SSP2-
4.5, SSP3-7.0 and SSP5-8.5 are available (21 [as of November 2019]; Supplementary Information Table S1; (Eyring et al.
2016; O'Neill et al. 2016)). A single ensemble member per model is used from the CMIP5 and CMIP6 archives. All simulations
are regridded conservatively to a regular 2.5°×2.5° grid.





**Table 1:** Single-Model Initial-Condition Large Ensembles (SMILEs) used in this study. Table reproduced from (Deser et al. in review). See also **http://www.cesm.ucar.edu/projects/community-projects/MMLEA/**

| Modeling center | Model version | Resolution (atm/ocn) | Years | Initialization (Methods) | # members | Forcing | Reference |
|---|---|---|---|---|---|---|---|
| CCCma | CanESM2 | ~2.8°x2.8°/~1.4°x0.9° | 1950-2100 | Macro and Micro | 50 | historical, rcp85 | (Kirchmeier-Young et al. 2017) |
| CSIRO | MK3.6 | ~1.9°x1.9°/~1.9°x1.0° | 1850-2100 | Macro | 30 | historical, rcp85 | (Jeffrey et al. 2013) |
| GFDL | ESM2M | 2.0°x2.5°/1.0°x0.9° | 1950-2100 | Macro | 30 | historical, rcp85 | (Rodgers et al. 2015) |
| GFDL | CM3 | 2.0°x2.5°/1.0°x0.9° | 1920-2100 | Micro | 20 | historical, rcp85 | (Sun et al. 2018) |
| MPI | MPI-ESM-LR | ~1.9°x1.9°/ nominal 1.5° | 1850-2099 | Macro | 100 | historical, rcp26, rcp45, rcp85 | (Maher et al. 2019) |
| NCAR | CESM1 | ~1.3°x0.9°/nominal 1.0° | 1920-2100 | Micro | 35* | historical, rcp85 | (Kay et al. 2015) |
| SMHI/KNMI | EC-EARTH | ~1.1°x1.1°/nominal 1.0° | 1860-2100 | Micro | 16 | historical, rcp85 | (Hazeleger et al. 2010) |

15    * CESM1: only the first 35 members of 40 available are used, since members 36-40 are slightly warmer (see http://www.cesm.ucar.edu/projects/community-projects/LENS/known-issues.html), which can affect the variability estimates when calculated across the ensemble.

## 2.2 Uncertainty partitioning



20   We partition three sources of uncertainty largely following HS09, such that the total uncertainty ($T$) is the sum of the model uncertainty ($M$), the internal variability uncertainty ($I$) and the scenario uncertainty ($S$), each of which can be estimated for a given time $t$ and location $l$:

$$T(t, l) = M(t, l) + I(t, l) + S(t, l)$$

with the fractional uncertainty from a given source calculated as $\frac{M(t,l)}{T(t,l)}$, $\frac{I(t,l)}{T(t,l)}$, and $\frac{S(t,l)}{T(t,l)}$.

There are different ways to define $M$, $I$ and $S$, in part depending on the information obtainable from the available model simulations (e.g., SMILEs versus CMIP). For the SMILEs, the model uncertainty $M$ is calculated as the variance across the ensemble means of the seven SMILEs (i.e., across the "forced responses" of the SMILEs). The ensemble means are additionally smoothed with a 10-year running mean for consistency with the HS09 approach discussed below, although the effect of this additional smoothing is minimal. The internal variability uncertainty $I$ is calculated as the variance across ensemble members of a given SMILE, yielding one estimate of $I$ per model. Prior to this calculation, time series are smoothed with a 10-year running mean corresponding to the target temporal resolution of decadal means. Averaging across the seven $I$ yields the multi-model mean internal variability uncertainty $I_{mean}$. Alternatively, to explore the assumption that $I_{mean}$ does not change over time, we use the 1950-2014 average value of $I_{mean}$ throughout the calculation (i.e., $I_{fixed}$). We also use the model with the largest and smallest $I$, i.e., $I_{max}$ and $I_{min}$, to quantify the influence of model uncertainty in the estimate of $I$.

The uncertainties $M$ and $I$ for CMIP, in turn, are calculated as in HS09: the forced response is estimated as a 4[th] order polynomial fit to the first ensemble member of each model, which is additionally smoothed with a 10-year running mean. The model uncertainty $M$ is then calculated as the variance across the estimated forced responses. To be comparable with the SMILE calculations, only simulations from RCP8.5 and SSP5-8.5 are used for the calculation of $M$ in CMIP; this neglects the fact that, for the same set of models, model uncertainty is typically slightly smaller in weaker emissions scenarios. The internal variability uncertainty $I$ is defined as the variance over time from 1950 to 2099 of the residual from the forced response of a given model. Prior to this calculation, time series are smoothed with a 10-year running mean, consistent with the calculations for SMILEs. Historical volcanic eruptions can thus affect $I$ in CMIP, while for SMILEs $I$ is more independent of volcanic eruptions since it is calculated across ensembles members. In practice, this difference was found to be very small (Supplementary Information Section S1). Averaging across all $I$ in CMIP yields the multi-model mean internal variability uncertainty $I_{mean}$, which, unlike the SMILE-based $I_{mean}$, is time-invariant. We also apply the HS09 approach to each ensemble member of each SMILE to explore the impact of the method choice.


Estimating the scenario uncertainty $S$ relies on the availability of an equal set of models that were run under divergent emissions scenarios. Since only few of the SMILEs were run with more than one emissions scenario, we turn to CMIP5 for the scenario





uncertainty. Following HS09, we calculate $S$ as the variance across the multi-model means calculated for the different emissions scenarios, using a consistent set of available models. We use the CMIP5-derived $S$ in all calculations related to

SMILEs. An alternative is to use the scenario uncertainty from a SMILE that provides ensembles for different scenarios (e.g., MPI-ESM-LR or the new CanESM5). The benefit would be a robust estimate of scenario uncertainty (since the forced response is well known), while the downside would be that a single SMILE is not representative of the scenario uncertainty as determined from multiple models (see Supplementary Information Section S2).

In addition to the fractional uncertainties, the total uncertainty for a multi-model multi-scenario mean projection is also calculated following HS09: 90% uncertainty ranges are calculated additively and symmetrically around the multi-model multi-scenario mean as $\pm \frac{1.654 \cdot \sqrt{I}}{F}$, $\pm \frac{1.654 \cdot (\sqrt{I} + \sqrt{M})}{F}$, and $\pm \frac{1.654 \cdot (\sqrt{I} + \sqrt{M} + \sqrt{S})}{F}$, with $F = \frac{\sqrt{I} + \sqrt{M} + \sqrt{S}}{\sqrt{I + M + S}}$. Note that the assumption of symmetry is an approximation, which is violated already by the skewed distribution of available emissions scenarios (e.g., 2.6, 4.5, and 8.5 Wm$^{-2}$ in CMIP5). Thus, the figures corresponding to this particular calculation should only be regarded as an illustration

rather than a quantitative depiction of the multi-model multi-scenario uncertainty. Also, the original depiction in HS09 was criticized for giving the impression of a "best estimate" projection resulting from averaging the responses across all scenarios. That impression is false since the scenarios are not assigned any probabilities, thus their average is not more likely to occur than any individual scenario. To avoid giving this false impression, here we rearrange the depiction of absolute uncertainty as compared to HS09.


## 3. Results

### 3.1 Global mean temperature and precipitation projection uncertainty

We first consider global area-averaged temperature and precipitation projections and their uncertainties (Figs. 1-2). Under RCP8.5 and SSP5-8.5, decadal global mean annual temperature is projected to increase robustly in the SMILEs and CMIP5/6

(Fig. 1a-c). Other scenarios result in less warming, as expected. These projections are then broken out by the different sources of uncertainties (Fig. 1d-f). Finally, the different uncertainties are expressed as time-evolving fraction of the total uncertainty, resulting in the iconic HS09 illustration (Fig. 1g-i). Note that Fig. 1d-f and Fig. 1g-i essentially show absolute and relative uncertainties. Thus, Fig. 1d-f is most useful to answer the question 'how large is the uncertainty of a projection for year X and what sources contribute how much?', while Fig. 1g-i is most useful to answer the question 'which sources are most important

to the projection uncertainty from now to year X?'. This nuance is easily appreciated when thinking about internal variability uncertainty, which remains roughly constant in an absolute sense, but approaches zero in a relative sense for longer lead times.

The projection uncertainty in decadal global mean temperature shows a familiar breakdown (Hawkins and Sutton 2009): internal variability uncertainty is important initially, followed by model uncertainty increasing and eventually dominating the



first half of the 21$^{st}$ century, before scenario uncertainty becomes dominant by about mid-century (Fig. 1g-i). SMILEs and
CMIP5 behave very similarly, attesting that the seven SMILE models are a good representation of the 28 CMIP5 models for
global mean temperature projections. CMIP6, in turn, shows a larger model uncertainty, both in an absolute (Fig. 1f) and
relative (Fig. 1i) sense. Since the scenario uncertainty in CMIP6 is by design similar to CMIP5 (spanning radiative forcings
from 2.6 to 8.5 Wm$^{-2}$), this result is indeed attributable to larger model uncertainty – consistent with the wider range of climate

sensitivities and transient responses reported for CMIP6 compared to CMIP5 (Tokarska et al. in review), a point we will return
to in Section 3.5. Absolute internal variability is slightly smaller in CMIP6 (Fig. 1f) compared to CMIP5, but not significantly
so, and therefore this factor is not responsible for the relatively smaller contribution to total uncertainty from internal variability
in CMIP6 (Fig. 1i).

Projections of global mean precipitation largely follow the breakdown found for temperature (Fig. 2). Again, SMILEs and
CMIP5 behave remarkably similarly, while CMIP6 shows larger model uncertainty compared to CMIP5; model uncertainty
dominates CMIP6 throughout the 21$^{st}$ century, still contributing >60% by the last decade (compared to ~45% in SMILEs and
CMIP5).



**Figure 1:** (a-c) 10-year running means of global annual mean temperature time series from (top) SMILEs, (middle) CMIP5, and (bottom) CMIP6, with observations (Rohde et al. 2013) superimposed in black, all relative to 1995-2014. For SMILEs, the ensemble mean of each model and the multi-model average of those ensemble means are shown; for CMIP the polynomial fit for each model and the multi-model average of those fits are shown. (d-f) Sources of uncertainty for the multi-model multi-scenario mean projection. (g-i) Fractional contribution of individual sources to total uncertainty. Scenario uncertainty for SMILEs in (g) is taken from CMIP5, since not all SMILEs offer simulations with multiple scenarios. (d-i) In all cases, the respective multi-model mean estimate of internal variability ($I_{mean}$) is used.





**Figure 2:** As Figure 1, but for precipitation (observations from Adler et al. 2003).

## 3.2 Spatial patterns of temperature and precipitation projection uncertainty

We recreate the maps from figure 6 in HS09 for decadal mean temperature, showing the spatial patterns of different sources of uncertainty for lead times of one, four, and eight decades, relative to the reference period 1995-2014 (Fig. 3). The patterns of fractional uncertainty contributions generally look similar for SMILEs and CMIP5/6 (and also similar to CMIP3 in HS09; not shown). In the first decade, internal variability contributes least in the tropics and most in the high latitudes. By the 4$^{th}$ decade, internal variability contributes least almost everywhere. Scenario uncertainty increases earliest in the tropics, where



signal-to-noise is known to be large for temperature (HS09; Mahlstein et al. 2011). By the 8[th] decade, scenario uncertainty dominates everywhere except over the subpolar North Atlantic and the Southern Ocean, owing to the documented model uncertainty in the magnitude of ocean heat uptake as a result of forced ocean circulation changes (Frölicher et al. 2015). While the patterns are largely consistent between the model generations (see also Maher et al., in review), there are differences in magnitude. As noted in Section 3.1, CMIP6 has a larger model uncertainty than CMIP5 (global averages for different lead times in CMIP6: 40%, 65%, 45%, in CMIP5: 14%, 26%, 24%). CMIP6 also has a longer consistent forcing period than CMIP5, as 'historical' forcing ends in 2005 in CMIP5 and 2014 in CMIP6. These two factors lead to the fractional contribution from scenario uncertainty being smaller in CMIP6 compared to CMIP5 and SMILEs throughout the century (global averages for different lead times in CMIP6: 2%, 26%, 54%, in CMIP5: 31%, 65%, 74%). Thus, the forcing trajectory and reference period need to be considered when interpreting uncertainty partitioning and when comparing model generations. An easy solution would be to ignore scenario uncertainty or normalize projections in another way (see Section 3.5).

The spatial patterns for precipitation generally also look similar between SMILEs and CMIP5/6 (and CMIP3 in HS11; Fig. 4). Internal variability dominates world-wide in the 1[st] decade, and remains important during the 4[th] decade, in particular in the extratropics, while the tropics start to be dominated by model uncertainty. The North Atlantic and Arctic also start to be dominated by model uncertainty by the 4[th] decade. Scenario uncertainty remains unimportant throughout the century in most places. While there is agreement on the patterns, there are notable differences between the SMILEs and CMIP5/6 with regard to the magnitude of a given uncertainty source: substantially more uncertainty gets partitioned towards model uncertainty in CMIP compared to SMILEs (global averages for 1[st] and 4[th] decade in CMIP5/6: 17/40% and 59/65%; in SMILEs: 9% and 34%) despite the good agreement between the global multi-model mean precipitation projections from SMILEs and CMIP (Fig. 2). Consequently, internal variability is smaller in CMIP than in SMILEs. This result is consistent with the expectation that, at small spatial scales (here 2.5°x2.5°), the HS09 polynomial fit tends to wrongly interpret internal variability as part of the forced response, thus artificially inflating model uncertainty (e.g., during the 4th decade; compare SMILEs and CMIP5). We quantify this "error" through the use of SMILEs in the next Section.





**Figure 3:** Fraction of variance explained by the three sources of uncertainty in projections of decadal mean temperature changes in 2015-2024, 2045-2054 and 2085-2094 relative to 1995-2014, from (a) SMILEs, (b) CMIP5 models, and (c) CMIP6 models. Percentage numbers give the area-weighted global average value for each map.

45





**Figure 4:** As in Fig. 3, but for precipitation.

50      **3.3 Role of choice of method to estimate the forced response**



One of the caveats of the HS09 approach is the necessity to estimate the forced response via a statistical fit to each model simulation rather than using the ensemble mean of a large ensemble. Here, we quantify the potential error that stems from using a 4th order polynomial to estimate the forced response in a perfect model setup. Specifically, we use one SMILE and treat each of its ensemble members as if it were a different model, applying the polynomial fit to estimate each ensemble member's forced response. By design, "model uncertainty" calculated from these forced response estimates should be zero (since they are all from a single model) and any deviation from zero will indicate the magnitude of the method error. We calculate this potential method error using each SMILE in turn. For global temperature, this error is small but clearly non-zero, and peaks around year 2020 at a contribution of about 10% to the total uncertainty (comprised of potential method error, internal variability and scenario uncertainty) and a range from 8-20% depending on which SMILE is used in the perfect model setup (Fig. 5a). The error decreases to <5% by 2040. For global precipitation, the error is larger, peaking at about 25% in the 2020s and taking until 2100 to reduce to <5% in all SMILEs (Fig. 5b). These potential errors are visible even in global mean quantities, where the spatial averaging should help in estimating the forced response from a single member. Consequently, potential errors are even larger at regional scales. For example, and to revisit some cases from HS09 and HS11, for decadal temperature averaged over the British Isles, the error contribution can range between 10% and 50% at its largest (Fig. 5c). For decadal monsoonal precipitation over the Sahel, the method error is also large and – due to the small scenario uncertainty and gradually diminishing internal variability contribution over time – contributes to the total uncertainty throughout the entire century (Fig. 5d).

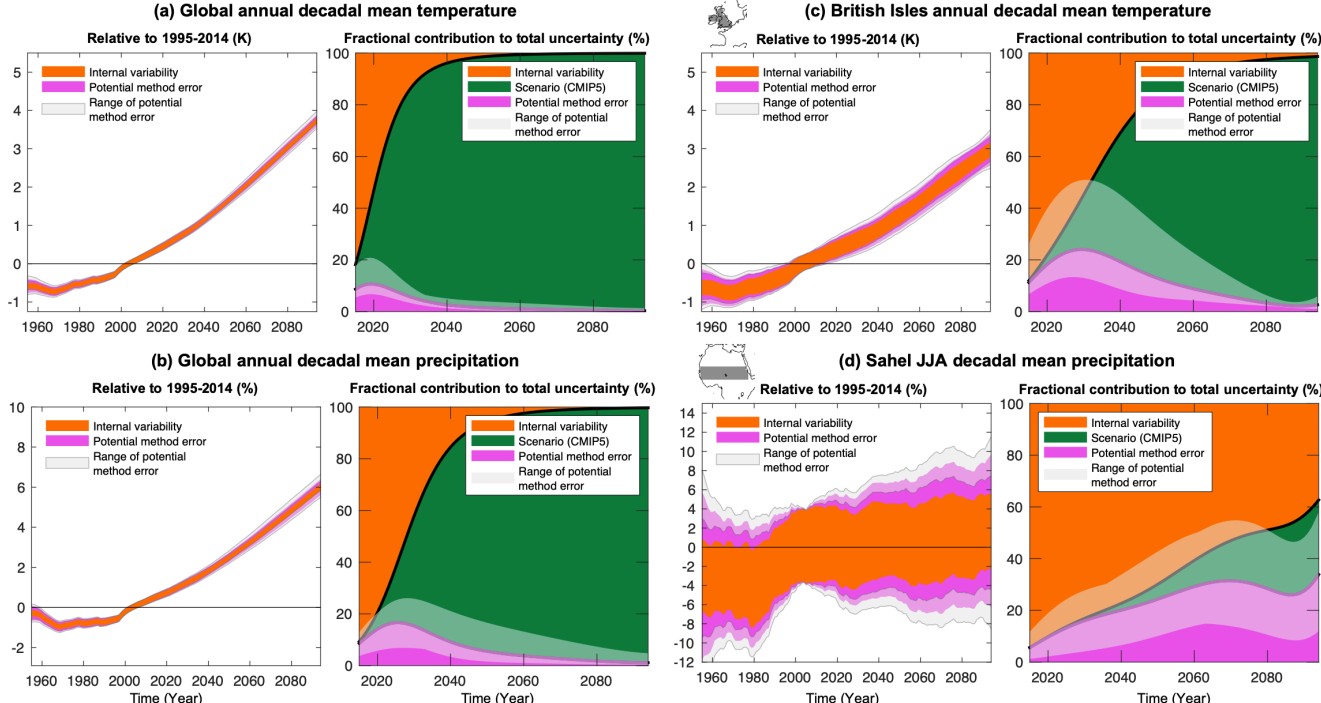

**Figure 5:** Decadal mean projections from SMILEs and fractional contribution to total uncertainty (using scenario uncertainty from CMIP5) for (a) global mean annual temperature, (b) global mean annual precipitation, (c) British Isles annual temperature, and (d) Sahel June-August precipitation. The pink color indicates the potential method error and is calculated the same way as model uncertainty in the HS09 approach, except instead of different models we only use different ensemble members from a SMILE, thus if the HS09 method were perfect, the error would be zero. This potential method error is calculated using each SMILE in turn and then the mean value from the seven SMILEs is used for the dark pink curve, while the slightly transparent white shading around the pink curve is the range of the potential method error based on different SMILEs.

The potential method error from using a polynomial fit has a spatial pattern, too (Fig. 6). For temperature, it is largest in the extratropics and smallest in the tropics (Fig. 6a). In regions of deep water formation, where the forced trend is small and an accurate estimate of it is thus difficult, the potential error contribution to the total uncertainty can be >50% even in the 4[th] decade. For precipitation, the potential method error is almost uniform across the globe and remains sizable throughout the century (Fig. 6b), consistent with the Sahel example in Fig. 5d. By the 8[th] decade, the contribution from potential method error starts to decrease and does so first in regions with a clear forced response (subtropical dry zones getting drier and high latitudes getting wetter), as there, scenario uncertainty ends up dominating the other uncertainty sources. The potential method error portrayed here can largely be reduced using SMILEs, at least if the ensemble size of a SMILE is large enough to robustly estimate the forced response (Coats and Mankin 2016; Milinski et al. 2019).

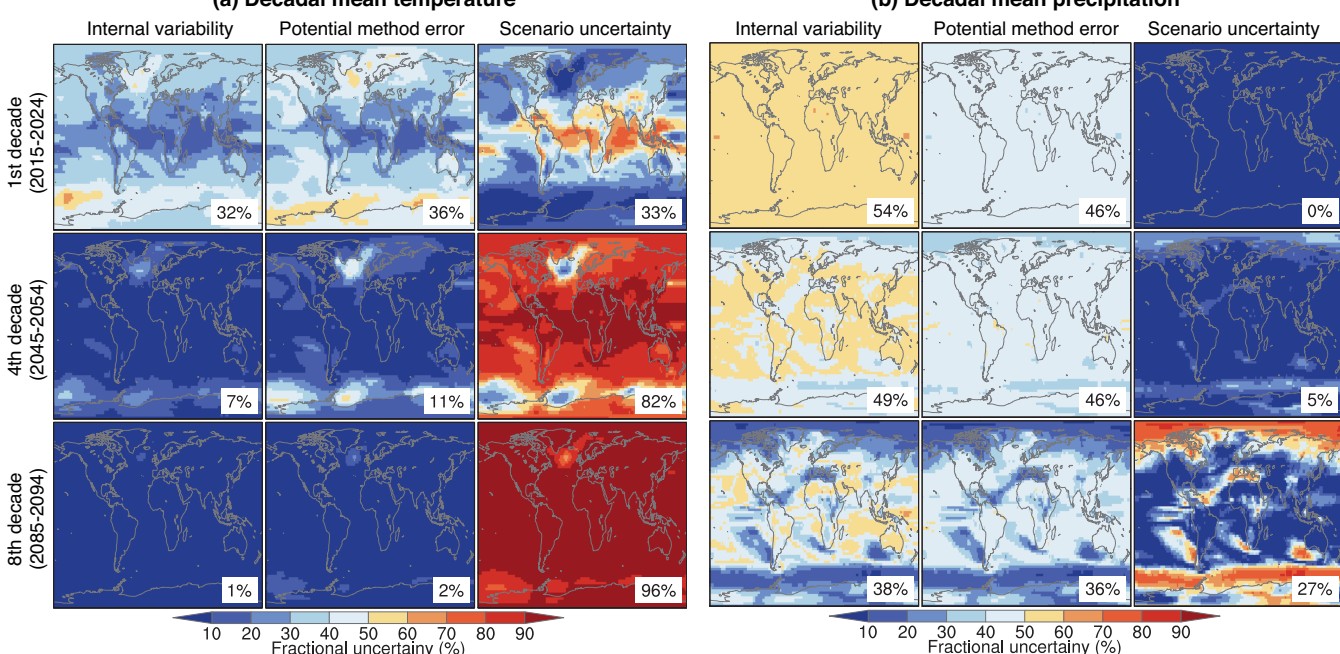

**Figure 6:** Fraction of variance explained by internal variability, potential method error, and scenario uncertainty in projections of decadal mean changes in 2015-2024, 2045-2054 and 2085-2094 relative to 1995-2014, for (a) temperature and (b) precipitation. The potential method error is calculated the same way as model uncertainty in the HS09 approach, except instead of different models we only use different ensemble members from one SMILE, thus if the HS09 method were perfect, the error would be zero. The potential method error is calculated using each SMILE in turn and then the mean value from the seven SMILEs is used for the maps here. Percentage numbers give the area-weighted global average value for each map.

If there are such large potential errors in estimating model uncertainty and internal variability, why are the results for SMILEs and CMIP5 still so similar (see Fig. 1 and 2)? Despite the imperfect separation of internal variability and forced response in HS09, the central estimate of variance across models is affected less if a large enough number of models is used (here, 28 from CMIP5). A sufficient number of models can partly compensate for the biased estimate of the forced response in any given model and – consistent with the central limit theorem – overall still results in a robust estimate of model uncertainty. The number of models needed varies with the question at hand and is larger for smaller spatial scales. For example, the potential method error for British Isles temperature appears to be too large to be overcome completely by the CMIP5 sample size, resulting in a biased uncertainty partitioning there (see also Section 3.5). HS09 used 15 CMIP3 models and large spatial scales to circumvent much of this issue, although it is important to remember that the potential error in estimating the variance of a population increases exponentially with decreasing sample size. In the special case of climate models, which can be interdependent (Masson and Knutti 2011; Knutti et al. 2013; Abramowitz et al. 2019), the potential error might grow slower or faster than that.



### 3.4 Role of model uncertainty in, and forced changes of, internal variability

Model uncertainty in internal variability itself can have an effect on some climate indices (Deser et al. in review; Maher et al. in review). The fraction of global temperature projection uncertainty attributable to internal variability varies by almost 50 percentage points around $I_{mean}$ at the beginning of the century, depending on whether $I_{max}$ or $I_{min}$ is used from the pool of SMILEs (range of white shading in Fig. 7a). This fraction diminishes rapidly with time as importance of internal variability generally decreases, but model differences in internal variability remain important over the next few decades (consistent with

Maher et al. in review). Global precipitation behaves similarly to temperature, except the range of internal variability contributions from the different SMILEs is smaller (Fig. 7b). Another example of uncertainty in internal variability itself is the magnitude of decadal variability of summer monsoon precipitation in the Sahel, which varies considerably across the SMILEs, resulting in internal variability contributing anywhere between about 40% and 80% in the first half of the century (range of white shading in Fig. 7c). The wide spread in the magnitude of variability across models suggests that at least some

models are biased in their variability magnitude. Understanding and resolving biases in variability in fully coupled models is important for attribution of observed variability as well as for efforts of decadal prediction. Sahel precipitation, for example, has a strong relationship with the Atlantic Ocean's decadal variability, which is one of the few predictable climate indices globally (Yeager et al. 2018); a realistic representation of variability would be one prerequisite for a skillful decadal prediction.

Internal variability can change in response to forcing, which can be assessed more robustly through the use of SMILEs. Comparing $I_{fixed}$ (which assumes no such change) with $I_{mean}$ shows that there is no clear forced change in decadal global annual temperature variability over time (Fig. 7a). Forced changes to precipitation variability are expected in many locations (Knutti and Sedláček 2012; Pendergrass et al. 2017), although robust quantification – in particular for decadal variability – has previously been hampered by the lack of large ensembles. Here, we show that forced changes in variability can now be detected

for noisy time series and small spatial scales, such as winter precipitation near Seattle, USA (Fig. 7d). Note, however, that in this example the increase in variability is small relative to the large internal variability, which is responsible for over 70% of projection uncertainty even at the end of the century. Forced changes in temperature variability are typically less wide-spread and less robust than those in precipitation, but can be detected in decadal temperature variability in some regions, for example the Southern Ocean (Fig. 7e). The projected decrease in temperature variability there could be related to diminished sea ice

cover in the future, akin to the Northern Hemisphere high latitude cryosphere signal (Screen 2014; Holmes et al. 2016; Brown et al. 2017), and around mid-century reduces the uncertainty contribution from internal variability by more than half compared to the case with fixed internal variability. Another example is the projected increase in summer temperature variability over parts of Europe (Fig. 7f; note that we have not applied the 10-year running mean to this example in order to highlight interannual variability), which is understood to arise from a future strengthening of land-atmosphere coupling (Seneviratne et

al. 2006; Fischer et al. 2012; Borodina et al. 2017).

**Figure 7:** Sources of uncertainty from SMILEs (using scenario uncertainty from CMIP5) for different regions, seasons and variables. The solid black lines indicate the borders between sources of uncertainty; the slightly transparent white shading around those lines is the range of this estimate based on different SMILEs. The dashed line marks the dividing line if internal variability is assumed to stay fixed at its 1950-2014 multi-SMILE mean. All panels are for decadal mean projections, except (f) Southern Europe Jun-Aug temperature, to which no decadal mean has been applied.

## 3.5 Uncertainties normalized by climate sensitivity

One of the emerging properties of the CMIP6 archive is the presence of models with higher climate sensitivity than in CMIP5 (Zelinka et al. in press; Tokarska et al. in review). As seen in Fig. 1 and 2, this can result in larger absolute and



relative model uncertainty for CMIP6 compared to SMILEs and CMIP5. However, it could be that this is merely a result of

the higher climate sensitivity and stronger transient response rather than indicative of increased uncertainty with regard to

processes controlled by (global) temperature. To understand whether this is the case, we express sources of uncertainties as a

function of global mean temperature (Fig. 8). For example, global mean precipitation scales approximately linearly with

global mean temperature under greenhouse gas forcing (Fig. 8a). Indeed, the absolute uncertainties from model differences

and internal variability are entirely consistent across SMILEs, CMIP5 and CMIP6 when normalized by global mean

temperature (Fig. 8b-c). Thus, uncertainty for global mean precipitation projections remains almost identical between the

different model generations, despite the seemingly larger uncertainty depicted in Fig. 2 for CMIP6. A counterexample is

projected temperature over the British Isles, where model uncertainty remains slightly larger in CMIP6 than in CMIP5 even

when normalized by global mean temperature (Fig. 8d-f). This example also illustrates once again the challenge of correctly

estimating the forced response from a single simulation, as the HS09 approach erroneously partitions a significantly larger

fraction of total uncertainty into model uncertainty compared to the SMILEs (Fig. 8b-c; see also Fig. 5b).





**Figure 8:** (a) Decadal means of global mean precipitation change as a function of global mean temperature change. Thin lines are forced response estimates from individual models, and thick lines are multi-model means for SMILEs, CMIP5 and CMIP6. The last decade of each multi-model mean is marked with a circle. (b) Uncertainty in global mean precipitation changes from model differences and internal variability for SMILEs, CMIP5 and CMIP6 as a function of global mean temperature. (c) Fractional contribution of global mean precipitation changes from model uncertainty and internal variability to total uncertainty as a function of global mean temperature. The colors indicate the fractional uncertainties from internal variability and model uncertainty in SMILEs, while the solid and dotted lines indicate where the dividing line between these two sources of uncertainty (i.e., between orange and blue colors) would lie for CMIP5 and CMIP6. (d-f) Same as (a-c) but for British Isles temperature.

Alternatively, models can be weighted according to performance metrics that are physically connected to their future warming magnitude. The original HS09 paper proposed using the global mean temperature trend over recent decades as an emergent constraint to determine if a model warms too much or too little in response to greenhouse gas forcing. This emergent constraint is relatively simple and more comprehensive ones have since been proposed (Steinacher and Joos 2016). However, the original idea has recently found renewed application to overcome the challenge of estimating the cooling magnitude from anthropogenic aerosols over the historical record (Jiménez-de-la-Cuesta and Mauritsen 2019; Tokarska et al. in review). Despite regional variations, the aerosol forcing has been approximately constant *globally* after the 1970s, such that the global temperature trend since then is more likely to resemble the response to other anthropogenic forcings, chiefly greenhouse gases (GHGs), which have steadily increased over the same time. Thus, this period can be used an observational constraint on the model sensitivity to GHGs. The correlation between the recent warming trend (1981-2014) and the longer trend projected for this century (1981-2100; using RCP8.5 and SSP5-8.5) is significant in CMIP5 (r=0.53) and CMIP6 (r=0.79), suggesting the existence of a meaningful relationship (Tokarska et al. in review). Following HS09, a weight $w_m$ can be calculated for each model:

$$w_m = \frac{1}{x_{obs}+|x_m-x_{obs}|},$$

with $x_{obs}$ and $x_m$ being the observed and model-simulated global mean temperature trend from 1981 to 2014. We apply the weighting to CMIP5 and CMIP6, but only to the data used to calculate model uncertainty – scenario uncertainty and internal variability remain unchanged for clarity. The weighting results in an initial reduction of absolute and relative model uncertainty for global mean temperature projections (Fig. 9). The reduction is larger for CMIP6 than for CMIP5, consistent with recent studies suggesting that CMIP6 models overestimate the response to GHGs (Tokarska et al. in review). Consequently, the weighting brings CMIP5 and CMIP6 global temperature projections into closer agreement, although remaining differences and questions, such as how aggressively to weigh models or how to deal with model interdependence (Knutti et al. 2017), are still to be understood.



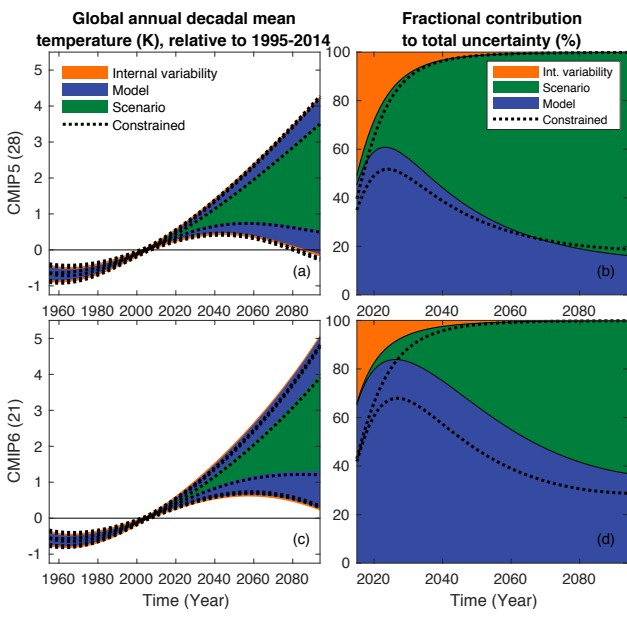

**Figure 9:** (a) Sources of uncertainty for the multi-model multi-scenario mean projection of global annual decadal mean temperature in CMIP5. (b) Fractional contribution of individual sources to total uncertainty. Observationally-constrained projections are given by the dotted lines (see text for details). (c-d) Same as (a-b) but for CMIP6.

## 4. Discussion and conclusions

We have assessed the projection uncertainty partitioning approach of Hawkins and Sutton (2009; HS09), in which a 4[th] order polynomial fit was used to estimate the forced response from a single model simulation. We made use of Single-Model Initial-Condition Large Ensembles (SMILEs) with seven different climate models (from the MMLEA) as well as the CMIP5/6 archives. The SMILEs facilitate a more robust separation of forced response and internal variability and thus provide an ideal testbed to benchmark the HS09 approach. We confirm that for averages over large spatial scales (such as global temperature and precipitation), the original HS09 approach provides a reasonably good estimate of the uncertainty partitioning, with potential method errors generally contributing less than 20% to the total uncertainty. However, for local scales and noisy targets (such as regional or grid-cell averages), the original approach can erroneously attribute internal variability to model uncertainty, with potential method errors at times reaching 50%. It is worth noting that a large number of models can partly compensate for this method error. Still, a key result of this study is the need for a robust estimate of the forced response. There are different ways to achieve this – utilizing the MMLEA as done here is one of them. Alternatively, new techniques to quantify and remove unforced variability from single simulations, such as dynamical adjustment or signal-to-noise maximization can be used (Wallace et al. 2012; Smoliak et al. 2015; Deser et al. 2016; Sippel et al. 2019; Wills et al. in review) and should provide an improvement over a polynomial fit.



Along with a better estimate of the forced response, SMILEs also enable estimating forced changes in variability if a sufficiently large ensemble is available (Milinski et al. 2019). While this study focused mainly on decadal means and thus decadal variability – showing wide-spread increases in precipitation variability and high latitude decreases in temperature variability – , changes in variability can be assessed at all time scales (Mearns et al. 1997; Pendergrass et al. 2017; Maher et al. 2018; Deser et al. in review; Milinski et al. 2019). Whether variability changes matter for impacts needs to be assessed on

a case-by-case basis. For example, changes in daily temperature variability can have a disproportionate effect on the tails and thus extreme events (Samset et al. 2019). However, there is a clear need to better validate model internal variability, as we found models to differ considerably in their magnitude of internal variability (consistent with Maher et al. in review), a topic that has so far received less attention (Deser et al. 2018; Simpson et al. 2018). SMILEs, in combination with observational large ensembles (McKinnon et al. 2017; McKinnon and Deser 2018), are opening the door for that.


SMILEs are still not widespread, running the risk of being non-representative of the "true" model diversity (see Abramowitz et al. 2019 for a review). Thus, to make inferences from SMILEs about the entire CMIP archive, it is necessary to test the representativeness of SMILEs. Fortunately, the seven SMILEs used here are found to be reasonably representative for several targets investigated, but a more systematic comparison is necessary before generalizing this conclusion. In any case, further

additions to the MMLEA will continue to increase the utility of that resource (Deser et al. in review).

Finally, we found that the seemingly larger absolute and relative model uncertainty in CMIP6 compared to CMIP5 can to some extent be reconciled by either normalizing projections by global mean temperature or by applying a simple model weighting scheme that targets the emerging high climate sensitivities in CMIP6, consistent with other studies (Jiménez-de-

la-Cuesta and Mauritsen 2019; Tokarska et al. in review). Constraining the model uncertainty in this way brings CMIP5 and CMIP6 into closer agreement, although differences remain that need to be understood. Regional and multi-variate weighting schemes show promise in aiding this effort (Knutti et al. 2017; Lorenz et al. 2018; Brunner et al. 2019). Improving the reliability of projections will thus remain a focal point of future climate research, with methods for robust uncertainty partitioning being an essential part of that effort.




**Acknowledgments**

We thank the members of the US CLIVAR Working Group on Large Ensembles for discussion. We acknowledge US CLIVAR for support of the Working Group on Large Ensembles, the Multi-Model Large Ensemble Archive (MMLEA), and the workshop on Large Ensemblesthat took place in Boulder in 2019; this paper benefited from all of these resources. We

550 acknowledge the World Climate Research Programme's Working Group on Coupled Modelling, which is responsible for CMIP, and we thank the climate modeling groups for producing and making available their model output to CMIP and the MMLEA. For CMIP the U.S. Department of Energy's Program for Climate Model Diagnosis and Intercomparison provides coordinating support and led development of software infrastructure in partnership with the Global Organization for Earth System Science Portals. The National Center for Atmospheric Research is sponsored by the U.S. National Science

555 Foundation. F.L. is supported by a Swiss NSF Ambizione Fellowship (Project PZ00P2_174128) and U.S. NSF AGS-0856145, Amendment 87. L.B. was supported by the EUCP project, funded by the European Commission through the Horizon 682 2020 Programme for Research and Innovation: Grant Agreement 776613.

**Competing interest**

560 The author declare no competing interests.

**Data and code availability**

CMIP data is available from PCMDI (https://pcmdi.llnl.gov/); the large ensembles are available from the MMLEA (http://www.cesm.ucar.edu/projects/community-projects/MMLEA/); the observational datasets are available from the

565 respective institutions; code for analysis and figures is available from F.L.



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
