# Peer review of "Partitioning climate projection uncertainty with multiple Large Ensembles and CMIP5/6"

_Earth System Dynamics, 2019_

## Short Comment (SC1) · 20 Feb 2020

Thanks for an interesting paper. It looks like figure 3c and figure 4c are the same. Could you please check that? If there is a mistake, please also check the numbers on lines 22, 25 and 35. They give the same numbers as the figures. Maybe you also need to check some of the conclusions.

Kind regards

Gustav

---

## Author Comment (AC1) · 20 Feb 2020

Dear Gustav,

Thank you for pointing out this mistake. The mistake was such that Fig. 3c (temperature) was accidentally repeated in Fig. 4c (precipitation). As such, Fig. 3 and its discussion in the text are correct, while Fig. 4c and its discussion in the text (line 35) need to be corrected. The conclusions are not affected by this revision.

We are providing the correct Fig. 4 and text here and will include it in the revised version.

Original text line 35: "...global averages for 1st and 4th decade in CMIP5/6: 17/40% and 59/65%;..."

[Figure]

Corrected text line 35: "...global averages for 1st and 4th decade in CMIP5/6: 17/17% and 59/59%;..."

With thanks and best regards,

Flavio Lehner
* * *
**Decadal mean precipitation**
**(a) SMILEs**

[Figure]

**Fig. 1.** Corrected Fig. 4. Only change is in panel (c).

---

## Referee Comment (RC1) · Auroop Ganguly (Referee) · 10 Mar 2020

The manuscript is well presented, scientifically rigorous, timely and in my estimation, certainly worthy of progressing towards rapid publication.

However, I do have a few relatively minor comments and suggestions:

1. In the introduction, while discussing the three sources of uncertainty, the authors discuss (a) uncertainty from internal unforced variability as well as (b) response uncertainty or model uncertainty. I would suggest the authors add at least a pointer (with a citation) to what has also been called "large ensemble of climate model simulations" which are obtained from "ensemble of model versions constructed by varying model parameters". One citation can be to Murphy et al. (2004), Nature 430, 768-772. How

this multi-parameter uncertainty fits into (or not) the overall discussion here could be useful for a reader to understand.

2. The skill versus consensus considerations when assigning uncertainties to projections may need to be discussed along with the need for physics consistency. Smith et al. (2009), Journal of the American Statistical Association, 104(485), pp.97-116, attempt to develop a statistical method for balancing skills versus consensus. An example of using physical basis for constraining uncertainty from models in provided in the context of precipitation extremes by O'Gorman and Schneider (2009), PNAS 106 (35) 14773-14777.

3. The Deser et al. (in review) paper is cited multiple times. Most journal allow authors to upload their manuscript on preprint servers such as arXiv without compromising novelty. Is that possible in this case?

4. Under uncertainty partitioning, the authors assume the three types of uncertainties are additive, without providing any context or caveats. The authors may need to clearly state whether this additive (and linearly separable) formulation for the three types of uncertainty is an assumption or a hypothesis, or if this is an assertion. In either case, the caveats of the assumption or a way to falsify the hypothesis or a rationale for the assertion should be provided.

5. While the manuscript makes a few distributional assumptions, I would suggest a more thorough discussion of a couple of points: first, the (assumed) asymptotic behavior (or the lack thereof) for the single-model initial condition ensembles (in other words, is there a reason to believe that with larger number of ensembles the distribution will converge or asymptote to a statistical distribution, and if so, is there any issue of ensemble sufficiency that needs to be investigated) and second, any assumption about the shape of the distributions (e.g., symmetric, or even Gaussian, etc.). I am not sure if there is enough basis to calculate the mathematical forms of the distributions from HS09.

6. The authors may need to discuss (and set appropriately in context) the visual intersection of the overall uncertainty with the zero-line in one case but not the other (Figure 1: (e) and (f)) and the relative changes in model and scenario uncertainty in CMIP5 versus CMIP6 (Figure 3: (b) and (c)).

7. The impact of internal variability on the phase difference of climate oscillators in model simulations may need to be discussed a bit more thoroughly. The others do provide an example with the Sahel. One example is provided in the context of the Indian monsoon is provided in Kodra et al. (2012), Environmental Research Letters 7(1): 014012, especially in the supplement.

8. As a stylistic matter, while I personally agree with the statement made by the authors that HS09 "created a powerful narrative of reducible and irreducible uncertainties in climate projections", I would nevertheless suggest deletion of superlatives such as "iconic framework" and "landmark paper". I would especially recommend this since one of the two authors of HS09 is also an author of this manuscript, but even otherwise.

9. As a "nice to have" suggestion, I am wondering if the value of this manuscript may be increased by a simplified exemplar. I am reminded of the ESM 2.0 paper: Schneider et al. (2017), Geophysical Research Letters, DOI: 10.1002/2017GL076101. That paper uses the Lorenz-96 model as an exemplar. While that GRL paper and this manuscript are very different indeed in scope and content, I was wondering if an exemplar such as a variant of the Lorenz model (with different initial conditions, different parameterizations as proxies for different "models", and a mock-up of different "forcing") may be developed here to clearly and concretely illustrate the basic points.

10. Finally, since most readers may not be familiar with HS09, I would suggest a clearer and more easily understandable (to a broad audience) discussion on what this means for the climate community, both for understand the science and for translating (or even starting to develop a conceptual framework to translate) to risk-informed decisions.

---

## Referee Comment (RC2) · Anonymous Referee #2 · 26 Mar 2020

This paper applies several large ensembles to evaluate to what extent the large ensembles can overcome limitations of an original approach by Hawkins and Sutton to quantify and illustrate uncertainty in future predictions and attribute it to different uncertainties. It is very well written and very interesting. I like that it also looks at temperature targets which is a nice extension of HS. i have two minor questions that the authors could consider strengthening - the conclusions mention the issue that the SMILEs might not quite span the model range and this kept worrying me through the paper - to what extent could differences between SMILEs and CMIP5/6 be due to a different set of models? ideally it would be nice to see a bit more discussion of this, or a sensitivity test to use a CMIP subset of runs that did SMILEs. (this might be near impossible with the original method of using only 1 run per model but that seems quite

a wasteful approach to me...). Also, do you think aerosol uncertainty is sampled well across SMILEs? Not doing so would have implications worth mentioning. - the point of changing variance of variability in the future (overcoming a particular limitation of HS) is a very interesting one and it doesnt get much space in the paper other than dashed lines in figure 7. could this be brought out a bit more - is the change in variance robust across models?

minor comments: the discussion of model imperfection l 44 or so could mention climate sensitivity uncertainty which also illustrates that even if technically reducable in practice this will be a slow processes. p 6 l 24: mention that I S M are variances Discussion l 17 p 21: This might also cross reference to signal to noise maximization used in optimal detection eg Hasselmann, 1979 or Allen and Tett.

---

## Author Comment (AC2) · 9 Apr 2020

The manuscript is well presented, scientifically rigorous, timely and in my estimation, certainly worthy of progressing towards rapid publication. However, I do have a few relatively minor comments and suggestions:

We thank the reviewer for the positive and constructive review. Our replies to the individual points raised are in blue.

1. In the introduction, while discussing the three sources of uncertainty, the authors discuss (a) uncertainty from internal unforced variability as well as (b) response uncertainty or model uncertainty. I would suggest the authors add at least a pointer (with a citation) to what has also been called "large ensemble of climate model simulations" which are obtained from "ensemble of model versions constructed by varying model parameters". One citation can be to Murphy et al. (2004), Nature 430, 768-772. How this multi-parameter uncertainty fits into (or not) the overall discussion here could be useful for a reader to understand.

Thanks for this useful reminder. As the reviewer is probably aware, this source of uncertainty is difficult to quantify for CMIP, but we completely agree that a pointer to its existence and its partial overlap with "model uncertainty" is needed here. To that end, the following paragraph is now included in the introduction:

- "Another important source of uncertainty not explicitly addressable within the CMIP context is parameter uncertainty. Even within a single model structure, considerable response uncertainty can result from varying model parameters in a perturbed-physics ensemble (Murphy et al. 2004; Sanderson et al. 2008). Such parameter uncertainty is sampled inherently but non-systematically through a set of different models, such as CMIP. Thus, it is currently convoluted with the structural uncertainty as described by "model uncertainty" and a proper quantification for CMIP is not possible due to the lack of perturbed-physics ensembles from different models."

2. The skill versus consensus considerations when assigning uncertainties to projections may need to be discussed along with the need for physics consistency. Smith et al. (2009), Journal of the American Statistical Association, 104(485), pp.97-116, attempt to develop a statistical method for balancing skills versus consensus. An example of using physical basis for constraining uncertainty from models is provided in the context of precipitation extremes by O'Gorman and Schneider (2009), PNAS 106(35) 14773-14777.

Thanks for raising this point. We have so far covered this very briefly in Section 3.5 and have now added a sentence in the Conclusions to stress this point more explicitly:

- "More generally, continued efforts are needed to include physical constraints when characterizing projection uncertainty, with the goal of striking the right balance between rewarding model skill, honouring model consensus, and guarding against model interdependence (Giorgi and Mearns 2002; Smith et al. 2009; O'Gorman and Schneider 2009; Sanderson et al. 2015)."

3. The Deser et al. (in review) paper is cited multiple times. Most journal allow authors to upload their manuscript on preprint servers such as arXiv without compromising novelty. Is that possible in this case?

Apologies for the inconvenience.

- Deser et al. (2020) is now published: https://www.nature.com/articles/s41558-020-0731-2
- Maher et al. (2020) is now published as well: https://iopscience.iop.org/article/10.1088/1748-9326/ab7d02

- The Wills et al. (in review) draft is accessible here:
  https://atmos.washington.edu/~rcwills/papers/2020_Wills_etal_Forced_Patterns.pdf

4. Under uncertainty partitioning, the authors assume the three types of uncertainties are additive, without providing any context or caveats. The authors may need to clearly state whether this additive (and linearly separable) formulation for the three types of uncertainty is an assumption or a hypothesis, or if this is an assertion. In either case, the caveats of the assumption or a way to falsify the hypothesis or a rationale for the assertion should be provided.

Indeed, this is an assumption and it is known to not be perfectly valid; for example, scenario and model uncertainty are not orthogonal, as discussed in, e.g., Yip et al. (2011). We agree that we did not emphasize this adequately. We have added a discussion of this caveat at the beginning of Section 2.2:

- "This formulation assumes the sources of uncertainty are additive, which strictly speaking is not valid due the terms not being orthogonal (e.g., model and scenario uncertainty). In practice, an ANOVA formulation with interaction terms results in similar results and conclusions (Yip et al. 2011)."

5. While the manuscript makes a few distributional assumptions, I would suggest a more thorough discussion of a couple of points: first, the (assumed) asymptotic behavior (or the lack thereof) for the single-model initial condition ensembles (in other words, is there a reason to believe that with larger number of ensembles the distribution will converge or asymptote to a statistical distribution, and if so, is there any issue of ensemble sufficiency that needs to be investigated) and second, any assumption about the shape of the distributions (e.g., symmetric, or even Gaussian, etc.). I am not sure if there is enough basis to calculate the mathematical forms of the distributions from HS09.

With regard to the first point: for the climate system the assumption is certainly that a sufficiently large ensemble converges to a statistical distribution and that this is very often a normal distribution. In fact, for the data in our study, the null hypothesis of a normal distribution cannot be rejected for large parts of the globe for both temperature and precipitation (not shown, but tested for the SMILEs), partly due to the temporal resolution (here annual or decadal means) and the coarse spatial resolution, but this is also consistent with more general expectations. All the regional examples discussed in the paper are normally distributed with high confidence (p<0.01; Shapiro-Wilk test) even when assessed across the ensemble dimension with the smallest ensemble size (n=16).

However, we have added a test of the ensemble size sufficiency in context of Fig. 5. Specifically, we subsampled the largest SMILE (MPI), selecting 16 ensemble members at random (corresponding to the smallest SMILE size, EC-EARTH) 100 times, to calculate 100 estimates of the forced response. Analogous to the current Fig. 5, the variance across those estimates quantifies the potential method bias when the forced response is calculated as an ensemble mean from a SMILE with only 16 ensemble members. It is shown that this too results in a non-zero method bias, but its magnitude is considerably smaller than with the HS09 approach shown in Fig. 5. This is now discussed in Section 3.3 and shown in supplementary Fig. S5. With ensemble sizes n>16 the bias decreases further, as expected (not shown).

[Figure]

*Fig. S5: Same as Fig. 5 in the main paper, except to estimate the potential method bias, we use different ensemble mean estimates from the same SMILE instead of different models and the 4th order polynomial. Specifically, we randomly select 16 members from the largest SMILE (MPI) to mimic the ensemble size of the smallest SMILE (EC-EARTH) and calculate the ensemble mean. We do this 100 times and calculate the variance across these ensemble means to be the potential method bias. Thus if the SMILE ensemble mean method were perfect, the bias would be zero. This bias here is also non-zero but substantially smaller than with the HS09 approach (see Fig. 5 in main text).*

With regard to the second point: we feel that the distributional assumptions are clearly stated in the last paragraph of Section 2.2, including a cautionary note that these assumptions are not perfectly valid in certain cases, for example due to the asymmetric distribution of forcing scenarios. Thanks to the robustness of variance as a metric, we still expect these distributional assumptions to be reasonable. We have added "…and possibly also by the distribution of models, which constitute an ensemble of opportunity rather than a particular statistical distribution (Tebaldi and Knutti 2007)".

6. The authors may need to discuss (and set appropriately in context) the visual intersection of the overall uncertainty with the zero-line in one case but not the other (Figure1: (e) and (f)) and the relative changes in model and scenario uncertainty in CMIP5 versus CMIP6 (Figure 3: (b) and (c)).

To clarify the first point, we added the following sentence to the first paragraph of Section 3.1:

- "The lack of high sensitivity models in CMIP5 compared to CMIP6 result in the 90% uncertainty range intersecting with zero in CMIP5 (Fig. 1e), but not CMIP6 (Fig. 1f)."

The second point is discussed in detail already in the first paragraph in Section 3.2.

7. The impact of internal variability on the phase difference of climate oscillators in model simulations may need to be discussed a bit more thoroughly. The others do provide an

example with the Sahel. One example is provided in the context of the Indian monsoon in Kodra et al. (2012), Environmental Research Letters7(1): 014012, especially in the supplement.

We considered this comment carefully, but are not entirely sure what the reviewer is referring to. We agree, and state in the paper, that the representation of internal variability varies widely across models. We stress the need to evaluate model internal variability more, but consider such an evaluation beyond the scope of this study. We illustrate the issue at the example of Sahel precipitation, among others (monsoonal precipitation over India looks very similar in that models disagree on internal variability; not shown). Beyond that, it appears that the discussion about the presence and adequacy of oscillations in model simulations is ongoing (see also Mann et al. (2020) for a recent discussion, albeit on the topic of global temperature). However, we have revised and expanded the paragraph about internal variability, including a citation of Kodra et al. (2012) to read:

- "The wide spread in the magnitude of variability across models suggests that at least some models are biased in their variability magnitude. Understanding and resolving biases in variability in fully coupled models is important for attribution of observed variability as well as for efforts of decadal prediction. Sahel precipitation, for example, has a strong relationship with the Atlantic Ocean's decadal variability, which is one of few predictable climate indices globally (Yeager et al. 2018). In case such decadal variability originates from an underlying oscillation, the SMILE-sampling of different oscillation phases contributes to ensemble spread and also complicates the evaluation of simulated internal variability with short observational records. Similar issues have been documented for the Indian monsoon (Kodra et al. 2012). Thus, a realistic representation of variability together with initialization on the correct phase of potential oscillations are prerequisites for skillful decadal predictions."

8. As a stylistic matter, while I personally agree with the statement made by the authors that HS09 "created a powerful narrative of reducible and irreducible uncertainties inclimate projections", I would nevertheless suggest deletion of superlatives such as"iconic framework" and "landmark paper". I would especially recommend this since one of the two authors of HS09 is also an author of this manuscript, but even otherwise.

In hindsight, we completely agree and have removed those statements. Thank you for pointing this out.

9. As a "nice to have" suggestion, I am wondering if the value of this manuscript may be increased by a simplified exemplar. I am reminded of the ESM 2.0 paper: Schneider et al. (2017), Geo physical Research Letters, DOI: 10.1002/2017GL076101. That paper uses the Lorenz-96 model as an exemplar. While that GRL paper and this manuscript are very different indeed in scope and content, I was wondering if an exemplar such as a variant of the Lorenz model (with different initial conditions, different parameterizations as proxies for different "models", and a mock-up of different "forcing") may be developed here to clearly and concretely illustrate the basic points.

We agree that this would be nice to have, but also clearly see it as beyond the scope of this study. Further, it is not immediately clear how the Lorenz-96 model would illustrate the basic points more clearly than what we have already provided. Specifically, in the Lorenz-96 model, "forcing" acts to amplify an oscillatory behavior, while in the CMIP framework it acts to change the base state, so the initial framing is a bit different. Of course, this could be reformulated, but that seems unnecessarily complicated. Rather, one can think of a much

simpler linear toy model for a climate variable (e.g., temperature from one model $T_m$) with one instantaneous response time scale:

$$T_m(t) = \sigma_m \mathcal{N}(t) + F(t) \cdot \kappa_m,$$
$$\sigma = \mathcal{N}(\bar{\sigma}, \sigma_\sigma),$$
$$\kappa = \mathcal{N}(\bar{\kappa}, \kappa_\sigma),$$

where a time-varying forcing $F$ (e.g., radiative forcing in W/m$^2$ from RCPs) is scaled by a model-specific sensitivity $\kappa_m$ and added to a model-specific random-normal time series $\sigma_m \mathcal{N}$. The parameters for both $\sigma$ and $\kappa$ are themselves drawn from normal distributions with means $\bar{\sigma}$ and $\bar{\kappa}$ and standard deviations $\sigma_\sigma$ and $\kappa_\sigma$. The model makes the aforementioned distributional assumptions, does not allow for forced changes in internal variability, has no interaction between scenario and model uncertainty, and ignores response time scales slower than instantaneous, but those are not critical caveats when conveying the basic points here.

The parameters means $\bar{\sigma}$, $\bar{\kappa}$, $\sigma_\sigma$ and $\kappa_\sigma$ can be chosen to mimic CMIP behavior for a given variable and spatial resolution (e.g., decadal mean global temperature, Fig. R1 top), and can then be varied to arbitrarily scale internal variability (Fig. R1 middle) and model uncertainty (Fig. R1 bottom). Further, the number of models and ensemble members can be varied to illustrate the performance of different methods of estimating the forced response, e.g., one can demonstrate again that when using a CMIP-sized pool of models to partition uncertainty in global mean temperature, the HS09 approach works almost as well as if we had a 100-member SMILE for each CMIP model (Fig. R1 top). In turn, if internal variability is an order of magnitude larger, the HS09 approach tends to wrongly attribute internal variability to model uncertainty (Fig. R1 middle).

Many more examples could be thought of, but we do not think they add much to the examples already shown in the paper based on actual climate model simulations. We have, however, included a reference to the work of Lorenz in the Introduction.

[Figure]

**Fig. R1:** *Toy model results. Forced response estimates from (left column) HS09 approach and (middle column) SMILEs approach. (Right column) Fractional contribution to total uncertainty. The top row shows an example that closely matches CMIP5 global mean temperature. The middle row shows an example with much larger internal variability, representing something like a single grid cell at high latitudes. The bottom row has the same parameters as the top row except about three-times-as-wide of a distribution of model sensitivities. Model uncertainty is always estimated from the RCP8.5 scenario (red). The chosen parameters are given in the panels.*

10. Finally, since most readers may not be familiar with HS09, I would suggest a clearer and more easily understandable (to a broad audience) discussion on what this means for the climate community, both for understand the science and for translating (or even starting to develop a conceptual framework to translate) to risk-informed decisions.

In the introduction, we now re-iterate and emphasize key points made in HS09 and elsewhere:

- "Such a separation helps identify where model uncertainty is large and thus where investments in model development and improvement might be most beneficial

(HS09). A robust quantification of projection uncertainty will also benefit multi-disciplinary climate change risk assessments, which often rely on quantified likelihoods from physical climate science (Sutton 2019; King et al. 2015)."

**References**

Giorgi, F., and L. O. Mearns, 2002: Calculation of average, uncertainty range, and reliability of regional climate changes from AOGCM simulations via the "Reliability Ensemble Averaging" (REA) method. *J. Clim.*, **15**, 1141–1158, https://doi.org/10.1175/1520-0442(2002)015<1141:COAURA>2.0.CO;2.

King, D., D. Schrag, Z. Dadi, Q. Ye, and A. Ghosh, 2015: *Climate change: a risk assessment*. 154 pp.

Kodra, E., S. Ghosh, and A. R. Ganguly, 2012: Evaluation of global climate models for Indian monsoon climatology. *Environ. Res. Lett.*, **7**, https://doi.org/10.1088/1748-9326/7/1/014012.

Mann, M. E., B. A. Steinman, and S. K. Miller, 2020: Absence of internal multidecadal and interdecadal oscillations in climate model simulations. *Nat. Commun.*, **11**, 1–9, https://doi.org/10.1038/s41467-019-13823-w.

O'Gorman, P. A., and T. Schneider, 2009: The physical basis for increases in precipitation extremes in simulations of 21st-century climate change. *Proc. Natl. Acad. Sci. U. S. A.*, **106**, 14773–14777, https://doi.org/10.1073/pnas.0907610106.

Sanderson, B. M., R. Knutti, P. Caldwell, B. M. Sanderson, R. Knutti, and P. Caldwell, 2015: A Representative Democracy to Reduce Interdependency in a Multimodel Ensemble. *J. Clim.*, **28**, 5171–5194, https://doi.org/10.1175/JCLI-D-14-00362.1.

Smith, R. L., C. Tebaldi, D. Nychka, and L. O. Mearns, 2009: Bayesian modeling of uncertainty in ensembles of climate models. *J. Am. Stat. Assoc.*, **104**, 97–116, https://doi.org/10.1198/jasa.2009.0007.

Sutton, R. T., 2019: Climate science needs to take risk assessment much more seriously. *Bull. Am. Meteorol. Soc.*, **100**, 1637–1642, https://doi.org/10.1175/BAMS-D-18-0280.1.

Tebaldi, C., and R. Knutti, 2007: The use of the multi-model ensemble in probabilistic climate projections. *Philos. Trans. R. Soc. A Math. Phys. Eng. Sci.*, https://doi.org/10.1098/rsta.2007.2076.

Yip, S., C. A. T. Ferro, D. B. Stephenson, and E. Hawkins, 2011: A Simple, coherent framework for partitioning uncertainty in climate predictions. *J. Clim.*, **24**, 4634–4643, https://doi.org/10.1175/2011JCLI4085.1.

---

## Author Comment (AC3) · 9 Apr 2020

This paper applies several large ensembles to evaluate to what extent the large en-sembles can overcome limitations of an original approach by Hawkins and Sutton to quantify and illustrate uncertainty in future predictions and attribute it to different uncertainties. It is very well written and very interesting. I like that it also looks at temperature targets which is a nice extension of HS. I have two minor questions that the authors could consider strengthening.

We thank the reviewer for the positive and constructive review. Our replies to the individual points raised are in blue.

The conclusions mention the issue that the SMILEs might not quite span the model range and this kept worrying me through the paper - to what extent could differences between SMILEs and CMIP5/6 be due to a different set of models? Ideally it would be nice to see a bit more discussion of this, or a sensitivity test to use a CMIP subset of runs that did SMILEs. (this might be near impossible with the original method of using only 1 run per model but that seems quite a wasteful approach to me...).

We now provide this sensitivity test for the variables and regions where the HS09 approach still works more or less robustly (Figs. R1-5; not included in the paper). The subsampled CMIP5 results show good correspondence with both the SMILEs and the full CMIP5 results, confirming that the specific SMILE models are a good representation of CMIP5 for the cases investigated here. Further, we provide an overview figure with time series for different variables and regions to demonstrate visually that SMILEs cover much of the range of CMIP5 (new Fig. S4).

Qualitatively, this conclusion is also supported by the fact that the SMILE models happen to be fairly independent of one another with regard to model structure, specifically the atmosphere and ocean components, something that might contribute to their documented relative independence when measured by temperature and precipitation projections (Knutti et al. 2013) or other climate fields (Sanderson et al. 2015).

We have added the following sentence to Section 3.1 in the paper:
- "This holds for other variables and large-scale regions subsequently investigated (Fig. S4), which is also consistent with the coincidental structural independence between the seven SMILEs (Knutti et al. 2013; Sanderson et al. 2015a)."

Also, do you think aerosol uncertainty is sampled well across SMILEs? Not doing so would have implications worth mentioning.

This is a good question, but difficult to answer given that aerosol-only simulations are rare. A literature search reveals that, among the SMILE models used here, GFDL-ESM2M and MPI-ESM are representative of weak aerosol forcing models in CMIP5, and GFDL-CM3 is representative of strong aerosol forcing models, with the other SMILE models, for which aerosol forcing estimates are available, falling in between (Forster et al. 2013; Rotstayn et al. 2015). We tentatively conclude that the SMILE models are not systematically biased in their sampling of aerosol forcing, but a more in-depth analysis might be needed to strengthen this conclusion. This is true in particular for the impact of aerosols on regional hydroclimate, which can be different from its global impact. We have added a sentence about this in the Conclusions:
- "For example, while the seven SMILEs used here cover the range of global aerosol forcing estimates in CMIP5 reasonably well (Forster et al. 2013; Rotstayn et al. 2015), their representativeness for questions of regional aerosol forcing remains to be investigated."

The point of changing variance of variability in the future (overcoming a particular limitation of HS) is a very interesting one and it doesnt get much space in the paper other than dashed

lines in figure 7. could this be brought out a bit more - is the change in variance robust across models?

While we agree that the topic of forced changes in internal variability is interesting, we see an in-depth analysis as beyond the scope of this already lengthy paper. We have, however, investigated the robustness of the forced changes in internal variability across SMILEs for the variables and regions in the paper and found that whenever there was a forced change detectable in the multi-SMILE mean (Fig. 7d-f), all 7 SMILEs agreed on the sign of change. We have added this statement to the main paper.

Minor comments:
the discussion of model imperfection l44 or so could mention climate sensitivity uncertainty which also illustrates that even if technically reducable in practice this will be a slow processes.

Climate sensitivity is implicit in model response uncertainty, but we nonetheless added an explicit mention of this point, including a citation of Roe and Baker (2007).

p6 l24: mention that I S M are variances
We have added this clarification.

Discussion l17 p21: This might also cross reference to signal to noise maximization used in optimal detection eg Hasselmann, 1979 or Allen and Tett.
We have added these references.

**References**

Forster, P. M., T. Andrews, P. Good, J. M. Gregory, L. S. Jackson, and M. Zelinka, 2013: Evaluating adjusted forcing and model spread for historical and future scenarios in the CMIP5 generation of climate models. *J. Geophys. Res. D Atmos.*, **118**, 1139–1150, https://doi.org/10.1002/jgrd.50174.

Knutti, R., D. Masson, and A. Gettelman, 2013: Climate model genealogy: Generation CMIP5 and how we got there. *Geophys. Res. Lett.*, **40**, 1194–1199, https://doi.org/10.1002/grl.50256.

Roe, G. H., and M. B. Baker, 2007: Why is climate sensitivity so unpredictable? *Science (80-. ).*, **318**, 629–632, https://doi.org/10.1126/science.1144735.

Rotstayn, L. D., M. A. Collier, D. T. Shindell, and O. Boucher, 2015: Why does aerosol forcing control historical global-mean surface temperature change in CMIP5 models? *J. Clim.*, **28**, 6608–6625, https://doi.org/10.1175/JCLI-D-14-00712.1.

Sanderson, B. M., R. Knutti, and P. Caldwell, 2015: Addressing interdependency in a multimodel ensemble by interpolation of model properties. *J. Clim.*, **28**, 5150–5170, https://doi.org/10.1175/JCLI-D-14-00361.1.

[Figure]

*Fig. R1: Projections of global decadal mean annual temperature from (top row) the SMILEs, (middle row) the 7 SMILE models from CMIP5, processed with the HS09 approach, (bottom row) the 28 CMIP5 models. Top and bottom row are identical to the corresponding rows in Fig. 1 in the main text. The similarities of the results across the rows is an indication of the representativeness of the 7 SMILE models for CMIP5.*

[Figure]

*Fig. R2: Same as Fig. R1 but for global decadal mean annual precipitation.*

[Figure]

*Fig. R3: Same as Fig. R1 but for British Isles decadal mean annual temperature.*

[Figure]

*Fig. R4: Same as Fig. R1 but for Southern Ocean decadal mean annual temperature.*

[Figure]

*Fig. R5: Same as Fig. R1 but for Southern Europe JJA temperature.*

---

## Author Comment (AC4) · 9 Apr 2020

The manuscript has been revised by addressing all points by the reviewers (see individual replies to reviewers). In addition, the manuscript has been revised for readability and to address the following issues:

- Fig. 4c and associate text was an accidental duplicate of Fig. 3c and has been corrected.

- It was erroneously reported that the SMILE ensemble means and HS09 polynomial fits were smoothed with an additional 10-year running mean. They were not. The results to not depend on this. The respective text was removed from Section 2.2

[Figure]

- "Method error" was changed to "method bias" throughout the paper as we recognize that the issue with the HS09 approach is systematic rather than random in that it tends to attribute internal variability to model uncertainty (but not vice versa).

- An additional method to calculate scenario uncertainty was brought to our attention and included in the Supplementary Information to illustrate the various possible approaches, but not used for the main paper.
* * *